# Effect of Light Intensity on the Growth and Antioxidant Activity of Sweet Basil and Lettuce

**DOI:** 10.3390/plants11131709

**Published:** 2022-06-28

**Authors:** Rūta Sutulienė, Kristina Laužikė, Tomas Pukas, Giedrė Samuolienė

**Affiliations:** 1Lithuanian Research Centre for Agriculture and Forestry, Institute of Horticulture, Kaunas Str. 30, LT-54333 Babtai, Lithuania; kristina.lauzike@lammc.lt (K.L.); giedre.samuoliene@lammc.lt (G.S.); 2Elektros Taupymo Sprendimai, Liepu Str. 15, LT-53290 Kaunas, Lithuania; tomas.pukas@ecolight.lt

**Keywords:** LED, fertilization, antioxidants, phenols, biomass

## Abstract

Light and nutrients are among the most important factors for sustained plant production in agriculture. As one of the goals of the European Green Deal strategy is to reduce energy consumption, greenhouse growers focus on high-value crop cultivation with less-energy-demanding growing systems. This study aimed to evaluate the effect of fertilization at different light intensities on the growth of lettuce and basil and the activity of the antioxidant system. Sweet basil (*Ocimum basilicum*, ‘Opal’) and lettuce (*Lactuca sativa*, ‘Nikolaj’) were grown in a greenhouse supplementing natural light (~80 µmol m^−2^ s^−1^) with lighting at two photon flux densities (150 and 250 µmol m^−2^ s^−1^), 16 h photoperiod, and 20/16 °C day/night temperature in May (Lithuania, 55°60′ N, 23°48′ E). In each light regime treatment, half of the plants were grown without additional fertilization; the other half were fertilized twice a week with a complex fertilizer (NPK 3-1-3). The results showed that the antioxidant activity of basil was most affected by 150 µmol m^−^^2^ s^−^^1^ PPFD lighting and the absence of fertilization. Altered antioxidant activity in lettuce in the presence of 250 µmol m^−^^2^ s^−^^1^ PPFD additional light intensity and fertilization resulted in higher morphological parameters.

## 1. Introduction

Considering environmental factors in agriculture, light and nutrients are the most important elements for productive plant growth. When estimating cultivation costs, it is best to grow under natural daylight; however, in the temperate zone, enough sunlight for vegetables and herbs growing in greenhouses is available only during a small part of the year [1]. The ability to control the parameters of the growing environment in closed-environment agriculture, especially the quality and quantity of artificial light, progressively increases the production of vegetables and herbs due to specific effects on the plant growth, yield, and metabolic processes [2]. The daily light integral (DLI) is a particularly useful and reliable greenhouse growing tool that can help identify the need for additional lighting and use the European Green Deal strategy to reach a point of optimal intensity or extend the photoperiod.

During autumn and winter, the light level is low, and the photoperiod is short due to the latitude of Lithuania and other northern countries. The DLI ranges from 45 to 50 mol m^−^^2^ d^−^^1^ in summer and, in contrast, from 0 to 5 mol m^−^^2^ d^−^^1^ in winter [3]. Due to the low penetration of light in greenhouses, additional light sources such as high-pressure sodium lamps and light-emitting diodes (LEDs) are used for supplemental lighting [4]. One of the goals of Europe’s Green Deal strategy is to reduce energy consumption but still maintain high-quality output [5]. LED lamps outperform high-pressure sodium lamps based on their low power consumption and ability to easily control light intensity [4,6,7,8]. LED lamps can reduce energy consumption for lighting by up to 40–70% compared to other light sources such as high-pressure sodium lamps [9,10].

Light intensity affects plant development, metabolism, and the activity of the antioxidant system. The leaf area, the amount of accumulated dry mass, and the total amount of phenols are strongly dependent on the light intensity [6,11,12,13]. To fulfill the requirements of plant growth and development, attention must be paid not only to lighting but also to balanced nutrients, as a lack of nitrogen, phosphorus, and potassium can reduce plant growth, development, photosynthesis, and leaf area. Restriction of other essential nutrients also has adverse effects on the physiological parameters of the plants [14,15,16,17]. A lack of nutrients can reduce the accumulation of phenolic compounds and decrease antioxidant activity [18,19], which would mean a deterioration in the quality of the vegetables.

Antioxidants are a group of compounds that can neutralize oxidation processes in cells by acting as reducers, free radical scavengers, and inhibitors of radical species and other prooxidants such as metals. Leafy herbs and vegetables are excellent sources of antioxidants such as polyphenols, carotenoids, and vitamins C, E [4,6]. In plants, such compounds perform different functions, including signal transduction, protection against insects and pathogens, and prevention of oxidative damage. They are also responsible for imparting color to the plants and improving the strength of the aroma and the taste of produce [6]. Plant antioxidants play an important role in the human diet by reducing the risk of cardiovascular and neurodegenerative diseases, reducing inflammation, reducing the harmful effects of reactive oxygen species, and slowing down aging [4].

Seeking to grow high-quality and nutritious plants without wasting energy and saving investments, it is important to find a balance between lighting and fertilization depending on regional conditions. For example, the lighting intensity of 50, 150, 300, and 600 μmol m^−2^s^−1^ resulted in an increase in basil height, leaf area, and fresh and dry weight grown in the growing chamber [20]. However, the highest intensity caused photoinhibition—the basil leaves were brittle and did not adhere to the stem. Pennisi et al. [21] found that the most suitable light intensity for basil growth in a controlled environment is 250 μmol m^−2^s^−1^, and an increase in chlorophyll content and fresh and dry weight was observed. However, no significant correlation was found between the increase in light intensity and antioxidant activity, phenol content, and flavonoids.

Several studies describe the effect of light intensity and fertilization on lettuce [22,23,24]. Fu et al. [22] showed that red–blue LED lighting intensities of 60, 140, and 220 μmol m^−2^ s^−1^ and nitrogen fertilization rates of 7, 15, and 23 mmol L^−1^ differently affected lettuce growth. They found that the combination of higher light intensity and lower nitrogen content improved photosynthesis and yields. In another study [23], lettuce was grown in a closed-type plant factory system using LED lighting with four different intensities 200, 230, 260, and 290 μmol m^−2^ s^−1^ and three different photoperiods 18/6 (1 cycle), 9/3 (2 cycles) or 6/2 (3 cycles) light/dark. The most suitable conditions for lettuce growth were 290 μmol m^−^^2^ s^−^^1^ at a 6/2 photoperiod or 230 μmol m^−2^ s^−1^ at a 18/6 photoperiod. In both cases, the anthocyanin content and fresh and dry weights were the highest. Moreover, Song et al. [24] have studied the effects of lettuce grown in different concentrations (¼, ½, and ¾) of hydroponic solutions at different red–blue LED light intensities of 150, 250, and 350 μmol m^−2^ s^−1^. The highest amounts of anthocyanins, polyphenols, flavonoids, and antioxidant activity based on FRAP and DPPH analyses were determined under 350 μmol m^−^^2^ s^−^^1^ and at the lowest concentration of the hydroponic solution.

The common trend shows that higher quality and yields of basil and lettuce tend to be grown in controlled-environment chambers with higher light intensity and less fertilizer, but it is not known whether the same results can be achieved in greenhouses where environmental conditions are not perfectly controlled. Thus, this study aimed to analyze the demand for lighting intensity and nutrient requirements for basil and lettuce growth and antioxidant system activity.

## 2. Results

Photosynthetic photon flux density (PPFD) had a significant positive effect on both basil and lettuce vegetative growth, regardless of fertilization (Table 1). Natural light intensity resulted in a smaller leaf area and a lower number of leaves in both plants. Furthermore, it was noticed that basil and lettuce formed more leaves, but the leaf area was smaller in fertilized plants under 150 µmol m^−2^ s^−1^ additional lightning compared to unfertilized plants. In addition, there was no difference in leaf number and leaf area regardless of whether plants were fertilized or not under 250 µmol m^−2^ s^−1^ PPFD.

The effect of PPFD was highly pronounced on basil stem diameter and plant elongation (Figure 1). Additional lighting of 150 and 250 µmol m^−2^ s^−1^ stimulated not only the formation of new leaves but also leave size and the total leaf area. Plants were significantly taller under 150 and 250 µmol m^−2^ s^−1^ compared with plants grown under natural light. Basil was up to 2.3–2.6 times taller without fertilizer and up to 40–69% taller with fertilizer compared to basil grown under natural light (Table 2).

The additional lighting also had a strong effect on the diameter of the basil stems, under 250 µmol m^−2^ s^−1^, the basil stems were up to twice as thick compared with basils under natural light (Table 2). This allowed the plant to grow steadily upward and not fall apart (Figure 1). Meanwhile, fertilization had no significant effect on basil stem diameter.

The effect of light intensity on the lettuce was pronounced—not only the size of the lettuce but also the color changed (Figure 2). Additional lighting stimulated the formation of new leaves, lettuce with additional lighting formed up to 36.8–63.2% (150 µmol m^−2^ s^−1^) and up to 84.2% (250 µmol m^−2^ s^−1^) more leaves compared to lettuce grown under natural light (Table 1). Correspondingly, the leaves formed in the conditions with additional lighting reached up to two times larger size compared to lettuce grown under natural light.

The photosynthetic photon flux density affected the average plant weight of both, basil, and lettuce. Meanwhile, fertilization affected only the average basil weight under natural light and, at 150 µmol m^−^^2^ s^−^^1^, had no significant effect on the weight of the lettuce (Table 3). Fertilization increased the average weight of basil grown under natural light up to 55%, while fertilization with additional lighting (150 µmol m^−^^2^ s^−^^1^) significantly reduced the average weight of the plant to 26%; however, under 250 µmol m^−^^2^ s^−^^1^, PPFD fertilization had no significant effect on average plant weight. The dry and fresh weight ratio (DW/FW) was significantly increased due to the maximum PPFD. In addition, basil DW/FW increased by 12% when fertilized and grown in natural light. With additional lighting of 250 µmol m^−2^ s^−1^ PPFD, DW/FW for both plants increased up to 11–14% compared to plants grown under natural light (Table 3).

Compared to fertilized plants, significantly higher DPPH, ABTS, and FRAP activity were found in basil leaves grown under PPFD 150 and 250 µmol m^−^^2^ s^−^^1^ (Figure 3). Fertilization had a significant effect on the antioxidant activity of basil, whereas PPFD did not have such an effect. The antioxidant activity under 250 µmol m^−^^2^ s^−^^1^ PPFD in unfertilized basils increased up to 87%—DDPH, 38%—ABTS, and 32%—FRAP, while 150 PPFD increased 46%—DDPH, 21%—ABTS, and 23%—FRAP compared to fertilized basils. However, lettuce fertilization, unlike photosynthetic photon flux density, did not have a significant effect on antioxidant activity (Figure 3). DPPH increased up to 2–3 times, ABTS up to 39–58%, and FRAP up to 16–21% under 150 and 250 µmol m^−^^2^ s^−^^1^ PPFD, respectively, compared to lettuce grown under natural light.

Fertilization, as well as additional lighting, had a significant effect on the total phenolic compounds (TPC) of basil. The TPC of unfertilized basils under 150 µmol m^−^^2^ s^−^^1^ PPFD increased up to 11%, while under 250 µmol m^−^^2^ s^−^^1^ and with additional lightning, TPC did not change compared to basils grown under natural light. Fertilization had a significant effect on TPC in all tested light conditions—the TPC of fertilized basils increased up to 7–28% (respectively, for 80–250 µmol m^−^^2^ s^−^^1^ PPFD) compared with unfertilized basils (Figure 4). Significantly higher amounts of anthocyanins were accumulated in basils under 250 µmol m^−^^2^ s^−^^1^ PPFD without fertilization and under natural light with fertilization (Figure 4).

In lettuce, fertilization significantly affected TPC only with additional lighting (Figure 4). However, a significantly higher TPC was found in lettuce grown under natural light and in fertilized lettuce under 150 µmol m^−^^2^ s^−^^1^ PPFD. A significantly lower concentration of total anthocyanins was accumulated in lettuce grown under natural light regardless of fertilization. Fertilization significantly increased the concentration of anthocyanins under additional lighting up to 38–40% (respectively, for 150 and 250 µmol m^−^^2^ s^−^^1^ PPFD) compared to unfertilized lettuce (Figure 4). Anthocyanins also increased with increasing light intensity, but under 150 µmol m^−^^2^ s^−^^1^ PPFD with fertilization, there were no significant differences from 250 µmol m^−^^2^ s^−^^1^ PPFD without fertilization.

The PCA scatterplot shows an average coordinate of the total phenolic compounds; FRAP, DPPH, and ABTS radical scavenging activity; DW/FW; anthocyanins; and biometric parameters in sweet basil ‘Opal’ when basil was treated (with/without fertilization and average light intensity during grow time). The main two factors, F1 and F2 of PCA, explained 73.86% of the total variance response in treatments (Figure 5). According to F1, splitting data by light intensity, the groups that differed most were those under 4.6 mol m^−2^ d^−1^ (DLI) and 14.4 mol m^−^^2^ d^−^^1^ (DLI) regardless of fertilization. Meanwhile, the split between 4.6 and 14.4 mol m^−^^2^ d^−^^1^ (DLI) fertilized basil was explained by both factors 1 and 2, but unfertilized basils were split between 4.6 and 14.4 mol m^−^^2^ d^−^^1^ (DLI) only by F1. Considering F1, basil grown at 8.6 mol m^−^^2^ d^−^^1^ (DLI) differed significantly based on whether it was fertilized or unfertilized. Based on these results, at lower light intensities, fertilization has a slight effect on the set of measured parameters, whereas, at higher intensities, fertilization has a greater effect on basil.

The PCA scatterplot for lettuce also shows an average coordinate of the total phenolic compounds; FRAP, DPPH, and ABTS radical scavenging activity; DW/FW; anthocyanins; and biometric parameters as in sweet basil ‘Opal’ when lettuce was treated (with/without fertilization and average light intensity during grow time). The main two factors, F1 and F2 of PCA explained 74.93% of the total variance response in treatments (Figure 6), which is similar to the result of basil PCA. Lettuce data showed similar tendencies as that of basils. According to F1, for data, also, split by light intensity, the groups that differed the most were under 4.6 mol m^−^^2^ d^−^^1^ (DLI) and 14.4 mol m^−^^2^ d^−^^1^ (DLI) regardless of fertilization. Meanwhile, the split between 4.6 and 14.4 mol m^−^^2^ d^−^^1^ (DLI) fertilized lettuce was only identifiable by one factor (F1). According to lettuce results, fertilization has an impact on the set of measured parameters regardless of light intensity.

## 3. Discussion

The photoperiod and the amount of light a plant receives per day are determinants of its nutrient quality and biomass formation [23,25,26]. The effect of light intensity on plants is clearly expressed in Figure 1 and Figure 2.

Leafy herbs and vegetables such as basil and lettuce are very popular crops among farmers because they are easy to grow, have a high yield index, are suitable for hydroponic and closed farming, and at the same time have a high margin for profitability. Considering our results, the intensity of light significantly affected the physiological processes of basil and the accumulation of bioactive compounds (Figure 3 and Figure 4). The daily light integral (DLI) is the most important for plants, and previous studies have shown that the optimal DLI for basil is about 12.9 mol m^−2^ d^−1^; this would be 224 µmol m^−2^ s^−1^ at 16 h photoperiod [26]. However, there have been reports in which the photoperiod did not significantly affect the growth and physiological properties of basil, but it is recognized that it is necessary to analyze the effects of the light intensity and photoperiod [27]. In addition, Kiferle et al. [28] emphasize that basils need a low concentration of nitrogen for optimal plant growth. However, there are studies that show that plant biomass increases with increasing fertilization (regardless of whether it is outdoor growing or greenhouse) [29,30]. Based on our results, fertilization did not affect the total leaf area, plant weight, or dry and fresh weight ratio (Table 1 and Table 3), but it affected the height and average leaf area of basil: under 150 µmol m^−2^ s^−1^ PPFD, fertilized basils formed smaller, but more numerous leaves compared to unfertilized basils.

Fertilizers can be used in small quantities and have a significant effect on plant quality, for example, lower amounts of fertilizer have had a significant effect on the accumulation of rosmarinic acid [28]. Our results showed that fertilized basil accumulated significantly more total phenolic compounds compared to unfertilized ones. However, there were no significant differences between fertilized basils under 150 and 250 µmol m^−2^ s^−1^ PPFD (Figure 4). It has also been reported that basil grown with less fertilizer accumulated more total phenols, especially increasing the content of rosmarinic and caffeic acid. Fertilization can have an adverse effect on the antioxidant system; the more the fertilizer, the lower the antioxidant activity [31,32]. Fertilization inhibited antioxidant activity only under additional lighting. DPPH, ABTS, and FRAP activity significantly decreased under 150 and 250 µmol m^−2^ s^−1^ PPFD with fertilization, meanwhile without additional lighting, the opposite effect was observed (Figure 3). The effects of fertilization on anthocyanins in basil are contradictory—some studies show no effect of fertilization on anthocyanins [31], while others show a significant increase in anthocyanins regarding the use of fertilizers [33]. According to our results, in low-light conditions (up to 150 µmol m^−2^ s^−1^ PPFD), fertilization significantly increased the concentration of anthocyanins in basils leaves, while under 250 µmol m^−2^ s^−1^ PPFD, fertilized basil was found to have a significantly lower concentration of anthocyanins than that in unfertilized basil (Figure 4).

Additional lighting may be used in the greenhouse to increase growth and reduce the time for growth, but it is needed only when the daily light integral is naturally low [34,35]. The most-used light intensities vary between 100 and 300 µmol m^−^^2^ s^−^^1^ PPFD [27,36,37]. In our experiment, the natural light reached about 80 µmol m^−^^2^ s^−^^1^; according to the studies mentioned above, this is an insufficient intensity for the cultivation of basil. Our results showed that additional lighting significantly increased basil’s growth and biomass (Table 1, Table 2 and Table 3). These results agree with Dou et al. [27], basil growth increased until 224–290 µmol m^−^^2^ s^−^^1^ PPFD, which means that it is economically unprofitable to use higher-intensity lighting. The optimal light intensity according to these researchers is up to 220 µmol m^−^^2^ s^−^^1^, wherein fresh and dry weights increased linearly with light intensity. Furthermore, the additional white LED lamps created favorable conditions for basil’s growth; even white LED showed the same or a higher positive effect on basil’s fresh mass and growth as monochromatic light [38]. Basil grew more and bigger leaves under additional lighting (Table 1), but a stem of sufficient diameter to keep the plant stable was achieved only using 250 µmol m^−^^2^ s^−^^1^ PPFD (Figure 1 and Table 2). Thus, with additional lighting, basil not only produced more biomass, but the plant remained stable and robust (Figure 1).

The effect of light intensity on the antioxidant activity of basil and total phenols varied depending on fertilization (Figure 3 and Figure 4). Low-intensity lighting may have different effects on different antioxidants, such as no effect on total phenolic compounds but an increase in DPPH scavenging radical activity or a decrease in FRAP [39]. According to Pennisi et al. [21], antioxidant capacity (FRAP) and total phenolic compounds increased with increasing light intensity up to 250 µmol m^−^ m^−^^2^ s^−^^1^ PPFD, but under 300 µmol m^−^^2^ s^−^^1^ PPFD, they began to decrease. Our results showed an increase in total phenols in fertilized basil grown under 150 µmol m^−^^2^ s^−^^1^ PPFD. However, no significant differences were found between the light intensities of 150 and 250 µmol m^−^^2^ s^−^^1^, whereas the total phenols content of unfertilized basil grown under 150 µmol m^−^^2^ s^−^^1^ was significantly higher but lower than that at 250 µmol m^−^^2^ s^−^^1^ PPFD compared to cultivation under natural light (Figure 4). Similar results were obtained on antioxidant activity, significantly higher antioxidant activity was found under 150 µmol m^−^^2^ s^−^^1^ PPFD (Figure 3).

As shown in Figure 2, morphological differences between exposure groups are visible. Lettuce was larger and thicker in high light and thinner in low light. Statistical analysis showed that, under natural light in the greenhouse, lettuce was significantly smaller both with and without fertilization. Exceptionally, it was observed that, on increasing the light intensity to 250 µmol m^−2^ s^−1^, lettuce’s morphological parameters significantly increased, but no difference in fertilization was found. The results are confirmed by another study that states that the largest lettuces were grown at 220 µmol ^−^^2^ s^−^^1^ light intensity and with the lowest fertilizer concentration [22]. This was due to the fact that the structure and physiology of plants are regulated by light, as the initial reaction of plants during photosynthesis is completely dependent on light conditions [40,41]. Low-light conditions inhibit plant growth and productivity by affecting gas exchange, and excessive light intensity adversely affects the photosynthetic apparatus [42]. In addition, a lack or excess of light causes changes in the anatomy of the leaf by altering the length of cells such as the palisade parenchyma and the spongy parenchyma [43]. Such changes in the plant can determine the thickness of the leaves of the plant.

The intensity of lighting and fertilization determines the quality of the lettuce. The levels of vitamin C, soluble sugar, soluble protein, and anthocyanin in lettuce were higher at 250–300 µmol m^−^^2^ s^−^^1^ light intensities [42]. This was confirmed by other researchers [44], who found that at the light intensity of 250 µmol m^−^^2^ s^−^^1^ and the lowest fertilization, vitamin C, soluble protein, sugar, and free amino acid was the highest. Our study complements these studies that lettuce grown under additional lightning 250 µmol m^−^^2^ s^−^^1^ exposure showed higher antioxidant activity (Figure 3) as per DPPH, ABTS, and FRAP analyses. However, no differences were observed between the plants grown under this condition whether they were fertilized or not. Fertilized plants showed (Figure 4) higher levels of total phenols only under additional lighting; the anthocyanin content in lettuce was the highest under the most intense lighting using fertilization. The researchers found [24,44,45] that the plant increased polyphenols and flavonoids in response to lower fertilizer concentrations. In general, the highest levels of anthocyanins, polyphenols, flavonoids, FRAP, and DPPH were observed at the lowest fertilizer concentrations.

## 4. Materials and Methods

Sweet basil (*Ocimum basilicum*, ‘Opal’) and lettuce (*Lactuca sativa*, ‘Nikolaj’) (‘Agrofirma SĖKLOS’, Vilnius, Lithuania) were grown in a greenhouse under three different lighting intensity conditions (in May, Lithuania: 55°60′ N, 23°48′ E). Photosynthetic photon flux density (PPFD) was measured using photometer–radiometer RF-100 (Sonopan, Białystok, Poland) every two hours on three days—sunny, overcast, and moderately cloudy in the greenhouse—and presented as an average. The DLI of natural lighting beside leaves of basil and lettuce was 4.6 mol m^−^^2^ d^−^^1^ (80 µmol m^−^^2^ s^−^^1^), the DLI with supplemental light-emitting diode (LED) lighting was 8.6 mol m^−^^2^ d^−^^1^ (80 + 70 = 150 µmol m^−^^2^ s^−^^1^), and 14.4 mol m^−^^2^ d^−^^1^ (80 + 170 = 250 µmol m^−^^2^ s^−^^1^). For supplemental light, white (4000K) Linas Industry 105 W (LI-EN5/3L/10000) lamps (Elektros taupymo sprendimai, Vilnius, Lithuania) were used. The average day/night temperature was 20 ± 3/16 ± 3 °C; data were measured throughout the experiment (Termio+ data logger, Lubawka, Poland). Basils and lettuce were seeded into pots (500 mL) containing a peat substrate (Terraerden, Latvia) with NPK (100–160; 110–180; 120–200 mg L^−1^) with microelements (mg L^−1^): Mn (0.0145), Cu (0.311), Mo (0.0351), B (0.0214), Zn (0.0455) and Fe (0.642) (pH 5.5–6.5). Plants were watered (RO-type of water, pH 6.8, electrical conductivity 0.01 S m^−1^) when needed, maintaining a similar substrate moisture. Half of the plants were grown without additional fertilization; the other half of the plants were fertilized with complex NPK 3-1-3 fertilizers (Terra Grow, Plagron, Netherlands). For this, 1 mL of fertilizer concentrate was diluted with 200 mL of water.

### 4.1. Morphological Parameters

Five representative basil and lettuce plants were randomized selected for the measurement of morphological parameters; some plants were removed from the selection due to a possible edge effect. Leaf area (cm^2^) was evaluated with a leaf area meter (AT Delta—T Device, Cambridge, UK). Stem diameter and plant height were additionally measured for basil (mm). The dry mass of plants was determined by drying them at +70 °C for 48 h (Venti cell 222, Medcenter Einrichtungen, Gräfeling, Germany) to constant weight.

### 4.2. Antioxidant Activity and Total Phenolic Content

Extracts were prepared by grinding 0.5 g of plant leaves with liquid nitrogen and diluting them with 5 mL of 80% methanol. Each of the three biological replicates consisted of at least three conjugated plants and was repeated in three analytical replicates. The antioxidant properties of basil and lettuce leaves were then evaluated.

#### 4.2.1. ABTS

The ABTS (2,2′-azino-bis (3-ethylbenzothiazoline-6-sulphonic acid) radical cation was obtained by incubating the 7 mM ABTS stock solution with 2.45 mM potassium persulfate (K_2_S_2_O_8_; final concentration) and allowing the mixture to stand in the dark at room temperature for 12–16 h before use [45]. Thereafter, 20 μL of the prepared sample was mixed with 290 μL of ABTS solution (ABTS stock solution was diluted 1:7), and the absorbance was measured after 11 min (plateau phase) at 734 nm (SPECTROstar Nano, BMG Labtech microplate reader, Ortenberg, Germany). The ABTS scavenging activity of basil and lettuce leaves extracts was calculated as the difference between the initial absorbance and after reacting for 10 min. A calibration curve was determined using Trolox (6-hydroxy-2,5,7,8-tetramethychroman-2-carboxylic acid; 97% purity; Sigma-Aldrich, Burlington, MA, USA) as an external standard with a range of concentrations from 0.1 to 0.8 mM (R^2^ = 0.99). It was expressed as ABTS µmol scavenged per 1 g of fresh weight (µmol g^−1^ FW).

#### 4.2.2. DPPH

For DPPH (2-diphenyl-1-picrylhydrazyl) assay, the 126.8 μM DPPH (100% purity; Sigma-Aldrich, Burlington, MA, USA) solution was prepared in methanol [46]. Subsequently, 290 μL of the DPPH solution was transferred to a test tube and mixed with 20 μL of the basil and lettuce leaves’ extract. The absorbance was scanned at 515 nm (SPECTROstar Nano, BMG Labtech microplate reader, Ortenberg, Germany) while reacting for 16 min. The free radical scavenging capacity was expressed as μmol of DPPH radicals scavenged per 1 g of fresh weight (µmol g^−1^ FW). A calibration curve was determined using Trolox (6-hydroxy-2,5,7,8-tetramethychroman-2-carboxylic acid; 97% purity; Sigma-Aldrich, Burlington, MA, USA) as an external standard with a range of concentrations from 0.1 to 0.6 mM (R^2^ = 0.99).

#### 4.2.3. FRAP

The FRAP method is based on reducing ferric ion (Fe^3+^) to ferrous ion (Fe^2+^). The fresh working solution was prepared by mixing 300 mM, pH 3.6 acetate buffer, 10 mM TPTZ (2,4,6-tripyridyl-s-triazine) solution in 40 mM HCl, and 20 mM FeCl_3_ × 6H_2_O at 10:1:1 (*v*/*v*/*v*) [47]. Subsequently, 20 µL of the sample was mixed with 290 μL of working solution and incubated in the dark for 30 min. Readings of the colored product (ferrous tripyridyl-triazine complex) were then taken at 593 nm with a SPECTROstar Nano BMG Labtech microplate reader (Ortenberg, Germany). A calibration curve was determined using Fe_2_(SO_4_)_3_ (iron (III) sulfate; 97% purity; Sigma-Aldrich, Burlington, MA, USA) as an external standard with a range of concentrations from 0.005 to 0.5 mM (R^2^ = 0.99). The antioxidant power is expressed as Fe^2+^ antioxidant capacity (Fe^2+^ µmol g^−1^ FW).

#### 4.2.4. Total Phenolic Content

The total content of phenolic compounds was determined as gallic acid equivalents. A 20 µL aliquot of the sample extract was mixed with 20 µL of 10% (*w/v*) Folin–Ciocalteu reagent and 160 µL of 1 M Na_2_CO_3_ solution [48]. After incubation for 20 min in the dark, the absorbance was measured at 765 nm (SPECTROstar Nano, BMG Labtech microplate reader, Ortenberg, Germany). The total phenolic compounds’ quantity in mg g^−1^ was calculated from the calibration curve of the gallic acid (0.01–0.1 mg mL^−1^, R^2^ = 0.99).

#### 4.2.5. Anthocyanin Content

Monomeric anthocyanin pigments reversibly change color at pH 4.5 and pH 1.0. The colored oxonium form exists at pH 1.0, and the colorless hemiketal form predominates at pH 4.5. The difference in absorbance of the pigments at 520 nm is proportional to the pigment concentration. Results are based on cyanidin-3-glucoside (molar extinction coefficient of 26,900 L cm^−1^ mol^−1^ and molecular weight of 449.2 g mol^−1^). Two buffers were prepared with different pH: pH 1 potassium chloride (0.025M), pH adjusted with HCl using a pH meter (Hanna, Woonsocket, RI, USA); and pH 4.5 buffer sodium acetate (0.4M), pH adjusted with acetic acid using a pH meter. Absorbances at 520 and 700 nm were measured using a SPECTROstar Nano BMG Labtech microplate reader (Ortenberg, Germany) after 20–50 min. The same proportions of extract and buffer were used, 40 µL of extract and 160 µL of buffer were added to the microplates. This method is described in detail by Lee et al. [49].

### 4.3. Statistical Analysis

MS Excel Version 2010 and XLStat 2020 Data Analysis and Statistical Solution for Microsoft Excel (Addinsoft, Paris, France) statistical software were used for data processing. Analysis of variance (ANOVA) was carried out along with Tukey multiple comparison test for statistical analyses, *p* ≤ 0.05, three biological replicates, conjugated sample (from five plants’ leaves) for biochemical analysis—three analytical replicates.

## 5. Conclusions

The results showed that the antioxidant activity of basil was most affected by 150 µmol m^−2^ s^−1^ PPFD lighting and the absence of fertilization, which led to higher morphological parameters of basil. Altered antioxidant activity in lettuce in the presence of 250 µmol m^−2^ s^−1^ PPFD additional light intensity and fertilization resulted in higher morphological parameters. In summarizing the results, the needs of each plant species need to be assessed to produce more and higher-quality yield while saving energy.

## Figures and Tables

**Figure 1 plants-11-01709-f001:**
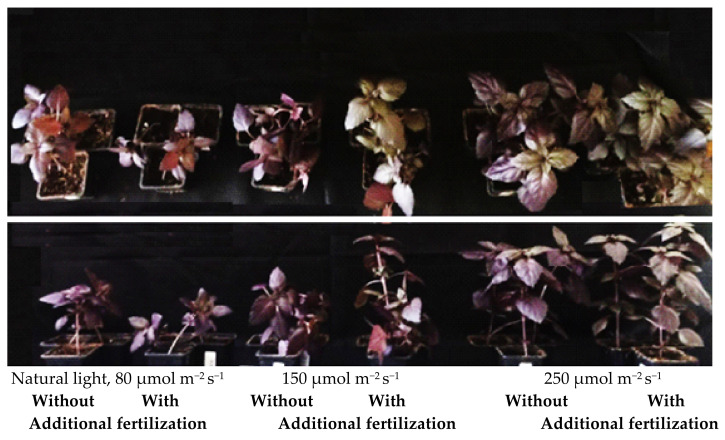
The effects of light intensity and fertilization on basil’s growth. Natural light corresponds to 4.6 mol m^−^^2^ d^−^^1^; 150 µmol m^−2^ s^−1^ corresponds to 8.6 mol m^−^^2^ d^−^^1^; 250 µmol m^−2^ s^−1^ corresponds to 14.4 mol m^−^^2^ d^−^^1^.

**Figure 2 plants-11-01709-f002:**
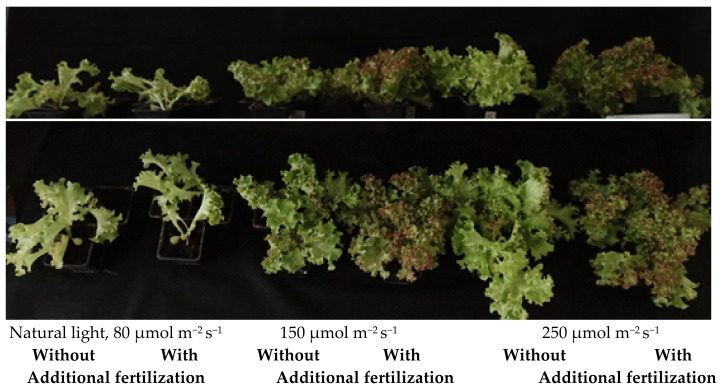
Light intensity and fertilization effect on lettuce growth. Natural light corresponds to 4.6 mol m^−^^2^ d^−^^1^; 150 µmol m^−2^ s^−1^ corresponds to 8.6 mol m^−^^2^ d^−^^1^; 250 µmol m^−2^ s^−1^ corresponds to 14.4 mol m^−^^2^ d^−^^1^.

**Figure 3 plants-11-01709-f003:**
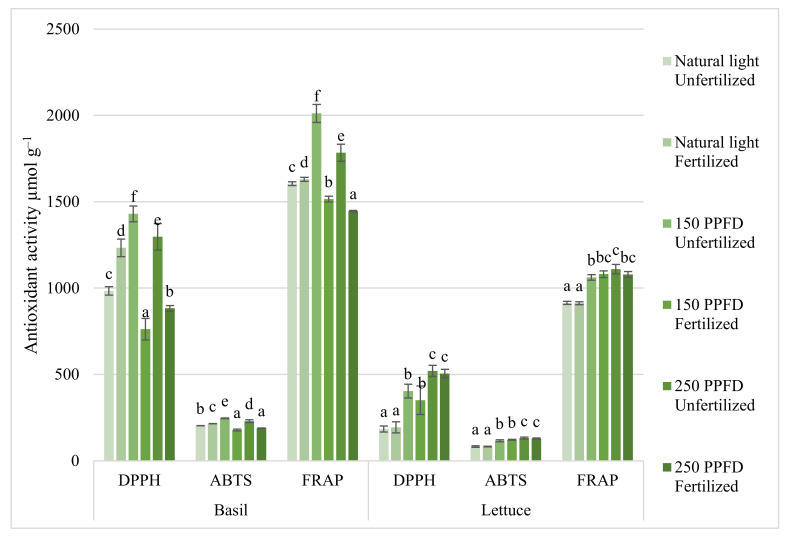
Effect of light intensity and fertilization on basil and lettuce antioxidant system activity. Natural light corresponds to 4.6 mol m^−^^2^ d^−^^1^; 150 µmol m^−^^2^ s^−^^1^corresponds to 8.6 mol m^−^^2^ d^−^^1^; 250 µmol m^−^^2^ s^−^^1^corresponds to 14.4 mol m^−^^2^ d^−^^1^. Values are mean ± SE of 10 replicates, and values with different letters differed significantly according to Tukey multiple comparison test (*p* ≤ 0.05).

**Figure 4 plants-11-01709-f004:**
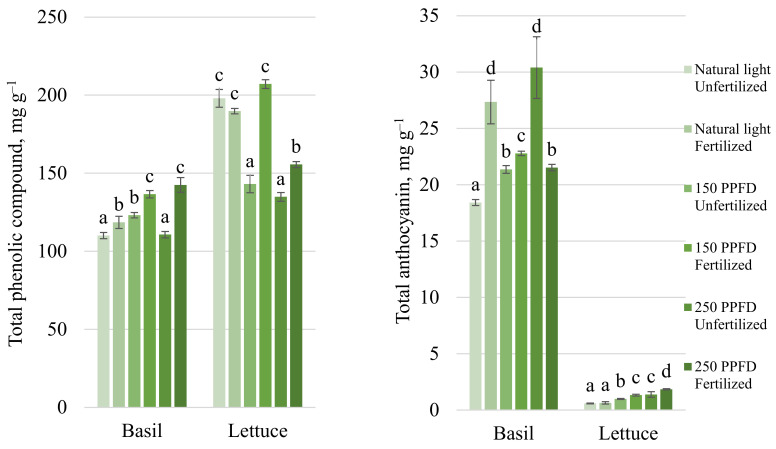
Effect of light intensity and fertilization on total phenolic compounds and anthocyanins in lettuce and basil. Natural light corresponds to 4.6 mol m^−^^2^ d^−^^1^; 150 µmol m^−^^2^ s^−^^1^ corresponds to 8.6 mol m^−^^2^ d^−^^1^; 250 µmol m^−^^2^ s^−^^1^ corresponds to 14.4 mol m^−^^2^ d^−^^1^. Values are mean ± SE of 10 replicates, and values with different letters differed significantly according to Tukey multiple comparison test (*p* ≤ 0.05).

**Figure 5 plants-11-01709-f005:**
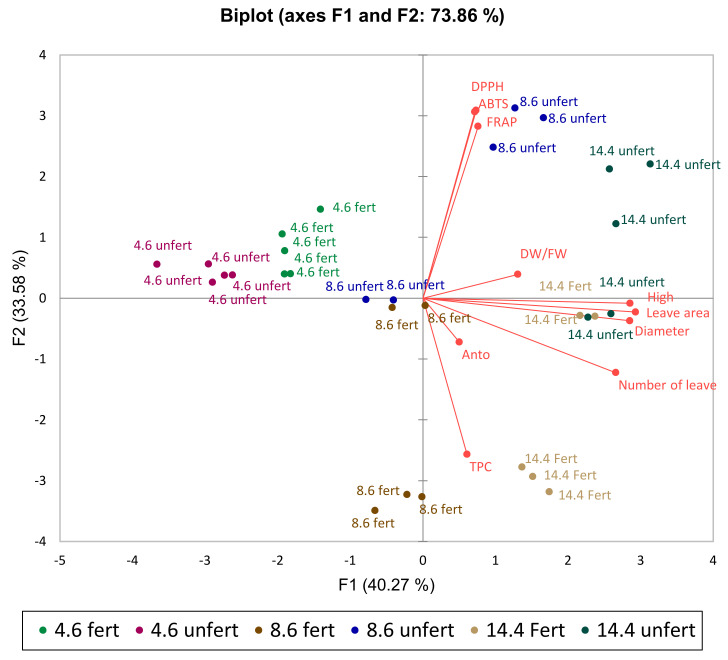
The principal component analysis (PCA) scatterplot indicates distinct differences in phenols; FRAP, DPPH, and ABTS radical scavenging activity; DW/FW; anthocyanins; and biometric parameters in sweet basil ‘Opal’. Natural light corresponds to 4.6 mol m^−^^2^ d^−^^1^; 150 µmol m^−^^2^ s^−^^1^ corresponds to 8.6 mol m^−^^2^ d^−^^1^; 250 µmol m^−^^2^ s^−^^1^ corresponds to 14.4 mol m^−^^2^ d^−^^1^.

**Figure 6 plants-11-01709-f006:**
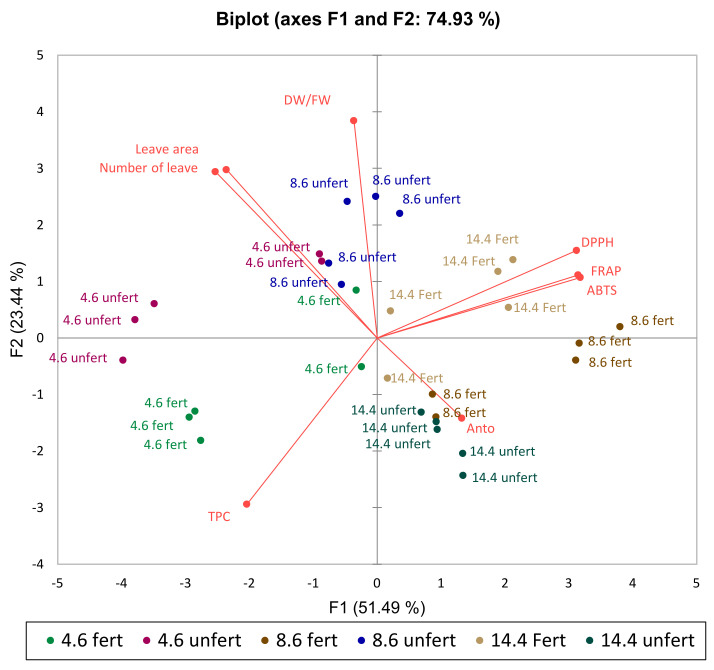
The principal component analysis (PCA) scatterplot indicates distinct differences in phenols; FRAP, DPPH, and ABTS radical scavenging activity; DW/FW; anthocyanins; and biometric parameters in lettuce ‘Nikolaj’. Natural light corresponds to 4.6 mol m^−^^2^ d^−^^1^; 150 µmol m^−2^ s^−1^ corresponds to 8.6 mol m^−^^2^ d^−^^1^; 250 µmol m^−2^ s^−1^ corresponds to 14.4 mol m^−^^2^ d^−^^1^.

**Table 1 plants-11-01709-t001:** The effect of light intensity and fertilization on the morphological parameters of basil and lettuce. Values are mean ± SE of 5 replicates, and values with different letters in columns differed significantly according to Tukey multiple comparison test (*p* ≤ 0.05).

Daily Light Integral (Photosynthetic Photon Flux Density)	Fertilizers	Total Leaf Area (cm^2^)		Number of Leaves		Average Leaf Area (cm^2^)	
		Basil	Lettuce	Basil	Lettuce	Basil	Lettuce
4.6 mol m^−^^2^ d^−^^1^ (natural light, 80 µmol m^−2^ s^−1^)	−	68.2 ± 5.35 a	128.3 ± 20.95 a	6 ± 0.0 a	3.8 ± 0.4 a	11.4 ± 0.89 a	33.8 ± 2.26 a
	+	113.4 ± 12.98 b	154.3 ± 36.85 a	6 ± 0.0 a	4.2 ± 0.4 a	18.9 ± 2.16 b	36.7 ± 5.45 a
8.6 mol m^−^^2^ d^−^^1^ (150 µmol m^−2^ s^−1^)	−	273.2 ± 50.58 c	363.5 ± 57.74 b	8 ± 0.0 b	5.2 ± 0.4 b	34.2 ± 6.32 c	69.9 ± 7.94 b
	+	214.8 ± 22.94 c	450.8 ± 30.47 bc	9.2 ± 0.4 c	6.2 ± 0.4 c	23.3 ± 1.34 b	72.7 ± 1.77 b
14.4 mol m^−^^2^ d^−^^1^ (250 µmol m^−2^ s^−1^)	−	409.1 ± 42.78 d	477.4 ± 41.50 c	10 ± 0.0 d	7 ± 0.0 d	40.9 ± 4.28 d	68.2 ± 5.93 b
	+	393.1 ± 28.32 d	516.9 ± 55.08 c	10 ± 0.0 d	7 ± 0.0 d	39.3 ± 2.83 d	73.8 ± 7.87 b

**Table 2 plants-11-01709-t002:** The effect of light intensity and fertilizer on basil high and stem diameter. Values are mean ± SE of 5 replicates, and values with different letters in columns differed significantly according to Tukey multiple comparison test (*p* ≤ 0.05).

Daily Light Integral (Photosynthetic Photon Flux Density)	Fertilizers	Basil Plant Height, cm	Basil Stem Diameter, mm
4.6 mol m^−^^2^ d^−^^1^ (natural light, 80 µmol m^−2^ s^−1^)	−	8.80 ± 2.1 a	1.98 ± 0.2 a
	+	12.48 ± 0.7 b	2.17 ± 0.1 a
8.6 mol m^−^^2^ d^−^^1^ (150 µmol m^−2^ s^−1^)	−	20.10 ± 2.8 cd	2.91 ± 0.4 ab
	+	17.48 ± 0.7 c	2.86 ± 0.2 b
14.4 mol m^−^^2^ d^−^^1^ (250 µmol m^−2^ s^−1^)	−	23.06 ± 0.6 e	4.05 ± 0.1 c
	+	21.20 ± 1.0 d	3.85 ± 0.1 c

**Table 3 plants-11-01709-t003:** Light intensity and fertilization effect on basil and lettuce plant weight and dry and fresh weight ratio. Values are mean ± SE of 5 replicates, and values with different letters in columns differed significantly according to Tukey multiple comparison test (*p* ≤ 0.05).

Daily Light Integral (Photosynthetic Photon Flux Density)	Fertilizers	Average Plant Weight, g		Dry/Fresh Weight Ratio, %	
		Basil	Lettuce	Basil	Lettuce
4.6 mol m^−^^2^ d^−^^1^ (natural light, 80 µmol m^−^^2^ s^−^^1^)	−	1.66 ± 0.13 a	2.57 ± 0.40 a	6.71 ± 0.05 a	6.16 ± 0.40 a
	+	2.57 ± 0.44 b	3.38 ± 0.56 a	7.54 ± 0.33 b	5.98 ± 0.27 a
8.6 mol m^−^^2^ d^−^^1^ (150 µmol m^−^^2^ s^−^^1^)	−	7.05 ± 0.65 d	11.49 ± 3.51 b	6.68 ± 0.08 a	6.74 ± 0.84 ab
	+	5.22 ± 0.25 c	15.86 ± 1.27 bc	6.75 ± 0.15 a	5.91 ± 0.73 a
14.4 mol m^−^^2^ d^−^^1^ (250 µmol m^−^^2^ s^−^^1^)	−	11.15 ± 0.75 e	17.20 ± 2.85 c	7.81 ± 0.22 b	7.02 ± 0.27 b
	+	10.85 ± 0.78 e	19.84 ± 3.18 c	7.50 ± 0.19 b	7.03 ± 0.31 b

## Data Availability

Not applicable.

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
