# Peer review of "Effect of Light Intensity on the Growth and Antioxidant Activity of Sweet Basil and Lettuce"

_plants, 2022, doi:10.3390/plants11131709_

Round 1

Reviewer 1 Report

This paper describes an experiment comparing growth parameters and antioxidant content of sweet basil and lettuce grown under 3 different lighting conditions and at two different nutrition levels. The research presented is relevant to the scientific community, however significant correction and upgrade of the manuscript is required prior to publication.

The description of the lighting conditions is not complete, it is not possible to reproduce the experiments based on the information provided in Chapter 4:

The natural light level was characterized by 80 μmol m−2s−1. It is not clear if this was a measured average PPFD value and how the natural light level and lighting duration were controlled. The term natural light might be misleading as in the study this corresponds to the lowest PPFD level. If natural light fluctuates over time, how can it be represented with a single number.

The specification of the supplemental LED lighting is not appropriate either: “For supplemental light white LED Tridonic 105W lamps (Elektros taupymo sprendimai, Vilnius, Lithuania) were used.” It would be useful to define “white” by correlated color temperature and publish the spectrum of the light source as a supplementary figure. It is not clear how to interpret 150 and 250 μmol m−2s−1. What is the distance between the light source and the plane in which PPFD was measured? Is the reported PPFD an average value? Is there any information on the spatial uniformity of PPFD? How was the PPFD level controlled?

From the text, the reader has the impression that 3 PPFD levels were applied which are the sum of the natural and LED illumination since LED was considered to supplement the natural light.

1: 80 μmol m−2s−1

2: 80 + 150 = 230 μmol m−2s−1

3: 80 + 250 = 330 μmol m−2s−1

 If this interpretation is correct, I propose to update related tables and figures to reflect the total amount of photon flux density.

The growth parameters vs. PPFD and fertilization are presented in Tables 1-3. It would be more informative to the reader to see scatter plots of the data. The authors miss the opportunity to highlight the linear relationship between total leaves area, fresh weight, etc., and PPFD.

There are several corrupted sentences that need to be corrected or rephrased:

Location

Issue

Proposal

Lines 19-21

The results showed that no additional fertilization was required, as the physiological response and intrinsic quality of basil and lettuce were statistically significantly higher compared to plants grown under natural light.

Logically inconsistent statement. Please rephrase.

Line 16.

natural with two additional lightning intensities

natural light with…

Line 32.

Daylight Integral (DLI)

Daily Light Integral

Line 37.

The average natural monthly light intensity ranges 37 from 45–50 mol·m–2·d–1 in summer

Monthly average daily light integral

Lines 42-43

LED lamps to surpass sodium lamps in their low power consumption and ability to easily control light intensity

Please rephrase

Lines 253-254

The researchers [44] found…

…confirmed by other researchers…

Please refer to the authors

Author Response

Response to Reviewer 1 Comments

We thank the reviewer for valuable comments on the manuscript. We are pleased to inform the reviewer that all suggestions have been considered and all requested edits have been made. The critical evaluation of the manuscript raised the following specific issues:

This paper describes an experiment comparing growth parameters and antioxidant content of sweet basil and lettuce grown under 3 different lighting conditions and at two different nutrition levels. The research presented is relevant to the scientific community, however significant correction and upgrade of the manuscript is required prior to publication.

The description of the lighting conditions is not complete, it is not possible to reproduce the experiments based on the information provided in Chapter 4:

Answer: Part of the methodology has been supplemented. See lines 272-289.

The natural light level was characterized by 80 μmol m−2s−1. It is not clear if this was a measured average PPFD value and how the natural light level and lighting duration were controlled. The term natural light might be misleading as in the study this corresponds to the lowest PPFD level. If natural light fluctuates over time, how can it be represented with a single number.

Answer:

What natural lighting is and what lighting measurements were made was also clarified, and the DLI was calculated. See in lines 274-279.

The specification of the supplemental LED lighting is not appropriate either: “For supplemental light white LED Tridonic 105W lamps (Elektros taupymo sprendimai, Vilnius, Lithuania) were used.” It would be useful to define “white” by correlated color temperature and publish the spectrum of the light source as a supplementary figure. It is not clear how to interpret 150 and 250 μmol m−2s−1. What is the distance between the light source and the plane in which PPFD was measured? Is the reported PPFD an average value? Is there any information on the spatial uniformity of PPFD? How was the PPFD level controlled?

White 4000K lamps with dimming function were used. We supplemented the medicine with information about lamps and PPFD measurements. The lamps were hung 80 cm from the plants, during the growth it was observed whether the lamps did not change the temperature between the plants, as no differences were found - we did not include this information in the manuscript. Light intensity was measured over the entire growth area in order to adjust the height of the lamps to the most uniform lighting conditions on the plant surface. Fluctuations in uniformity are less than 10 μmol m − 2s − 1 at plant height

 From the text, the reader has the impression that 3 PPFD levels were applied which are the sum of the natural and LED illumination since LED was considered to supplement the natural light.

1: 80 μmol m−2s−1

2: 80 + 150 = 230 μmol m−2s−1

3: 80 + 250 = 330 μmol m−2s−1

 If this interpretation is correct, I propose to update related tables and figures to reflect the total amount of photon flux density.

Answer:

We apologize for the incorrect description. Corrected in lines 277-279. It should have been:

1: 80 μmol m−2s−1

2: 80 + 70 = 150 μmol m−2s−1

3: 80 + 170 = 250 μmol m−2s−1

The growth parameters vs. PPFD and fertilization are presented in Tables 1-3. It would be more informative to the reader to see scatter plots of the data. The authors miss the opportunity to highlight the linear relationship between total leaves area, fresh weight, etc., and PPFD.

A PCA scatter plot was performed, and the data and their relationships and effects were analysed

There are several corrupted sentences that need to be corrected or rephrased:

Location

Issue

Proposal

Changes

Lines 19-21

The results showed that no additional fertilization was required, as the physiological response and intrinsic quality of basil and lettuce were statistically significantly higher compared to plants grown under natural light.

Logically inconsistent statement. Please rephrase.

The results showed that the antioxidant activity of basil was most affected by 150 µmol m−2s−1 PPFD lighting and the absence of fertilization.

Altered antioxidant activity in lettuce in the presence of 250 µmol m−2s−1 PPFD additional light intensity and fertilization resulted in higher morphological parameters.

Line 16.

natural with two additional lightning intensities

natural light with…

supplementing natural light (~80 µmol m-2 s-1) with two additional lightning intensities (150 and 250 µmol m-2 s-1)

Line 32.

Daylight Integral (DLI)

Daily Light Integral

Changed.

Line 37.

The average natural monthly light intensity ranges 37 from 45–50 mol·m–2·d–1 in summer

Monthly average daily light integral

Changed.

Lines 42-43

LED lamps to surpass sodium lamps in their low power consumption and ability to easily control light intensity

Please rephrase

LED lamps outperform sodium lamps in their low power consumption and easy control of light intensity

Lines 253-254

The researchers [44] found…

…confirmed by other researchers…

Please refer to the authors

LED lamps outperform high-pressure sodium lamps based on their low power consumption and ability to easily control light intensity

Reviewer 2 Report

Overall, this manuscript has scientific soundness with a clear presentation, however, I have a few questions below: 

1. its good to have one more treatment added like 350 micromole and see how the pattern changes from 250 mm. 

2. It should have a PCA data,  showing correlation with fertilizer and light intensity 

3. need a clear explanation, of how plant shows good activity even under fertilizer with high light? generally, the plant demands more fertilizer when grown under high light during TCA cycle. 

4. is scientifically sound to make a conclusion that, basil and lettuce require similar light intensity? 

5. Since, one of the objectives of this paper is energy reduction; however, providing 250 mmol light over 150 shows good results, so how is this design helps to reduce energy costs? 

6. is this experimental greenhouse completely controlled? or there was a small chamber within the greenhouse?if not controlled , how was the temperature managed? please describe details in M/M sections. 

Author Response

Response to Reviewer 2 Comments

We thank the reviewer for valuable comments on the manuscript. We are pleased to inform the reviewer that all suggestions have been considered and all requested edits have been made. The critical evaluation of the manuscript raised the following specific issues:

  1. its good to have one more treatment added like 350 micromole and see how the pattern changes from 250 mm. 

Answer: Based on the findings of previous researchers, we decided not to increase the lighting to 350 and more: Larsen et al., 2020; Pennisi et al., 2020; Fu et al., 2017; Kang et al, 2013, also, higher light intensity increase the cost of yields.

  1. It should have a PCA data, showing correlation with fertilizer and light intensity 

A PCA scatter plot was performed, and the data and their relationships and effects were analysed

  1. need a clear explanation, of how plant shows good activity even under fertilizer with high light? generally, the plant demands more fertilizer when grown under high light during TCA cycle. 

Answer: Primary results of our experiment show that fertilization increases the antioxidant activity in lettuce. Thank you for your suggestion, we agree that it would be very helpful to study the effects of light intensity along with increasing the fertilizer rate.

  1. is scientifically sound to make a conclusion that, basil and lettuce require similar light intensity? 

Answer: Thank you for the note, corrected in conclusion section.

  1. Since, one of the objectives of this paper is energy reduction; however, providing 250 mmol light over 150 shows good results, so how is this design helps to reduce energy costs? 

Answer: In terms of energy savings, we had in mind both the choice of lighting (LED's) and the use of fertilizers (their production also costs a lot of energy, we wanted to check whether it is worth using fertilizers if it is enough to maintain adapted lighting to achieve the same yield.

  1. is this experimental greenhouse completely controlled? or there was a small chamber within the greenhouse? if not controlled, how was the temperature managed? please describe details in M/M sections. 

Answer: Yes, the greenhouse was not controlled, changes were made to the methodological part, we provided an average of the conditions under which the plants grew.

Reviewer 3 Report

This is rather simple experiment, but it has been properly planned and performed, and interesting results are obtained. However, relatively large number of improvements are necessary in all parts of the manuscript.

Title

Use "activity" instead of "response". 

Do not use "antioxidant response" also in other places of the manuscript.

Abstract

Abstract is unbalanced, as results are mentioned only in a form of a short conclusion.

Rephrase the first sentence, as, for example, "Light and nutrients are among the most important factors for sustained plant production in agriculture".

Line 11, use "the European Green Seal strategy". Start the sentence with "As" and finish with using "less energy-demanding growing systems".

Line 14, use "activity" instead of "response".

Line 16, use "supplementing natural light with lighting at two photon flux densities".

Line 17–18, use "in each light regime".

Line 19, use "with a complex fertilizer".

Line 19–21, instead of a last sentence, describe the main results in two sentences and give a conclusion in a third.

Introduction

Lines 25–26, why the third factor is added to the two mentioned in the Abstract? It is rather confusing. Use the same construction as in the Abstract. 

Line 30, do not introduce abbreviation CEA, it is not further used.

Line 32–35, it is not becoming clear why the concept of daylight integral (or rather daily light integral) was introduced, as it is not included in the results of this study. No reference is given.

Lines 36–37, to avoid the impression that the study is important only for Lithuania, mention northern regions instead. Here, it is not light intensity, what is given, but rather daily light integral levels.

Line 41, use "the European Green Deal strategy".

Line 46–47, addressing "activity of the antioxidant system" (as an aspect of vegetable quality) needs to be explained in detail in a separate paragraph after describing light and nutrient effects on plant growth and production. 

Line 48–50, this thought is rather trivial.

Line 50–51, why only nitrogen and phosphorus are stressed out, there are number of other elements essential for plant growth?

Line 56–57, there is no confirmation in a form of a reference that basil and lettuce are the most popular vegetables in the world. It is advised to omit specific details in this and the following paragraph on basil and lettuce cultivation and to transfer these to discussion. Instead, give general information on choice of model species and the main characteristics in respect to demands for light and nutrients, and describe general scientific problem on need to balance additional light and fertilization in respect to natural light conditions of the particular region for particular crop species.

Results

In several places, the present tense is used instead of the past tense, consider changing all to the past tense (lines 107, 115).

Line 87, use "additional lighting" instead of "PPFD". Also, in other places, try to use "increasing PPFD" instead of simple "PPFD".

Line 95, use "morphological parameters" instead of "biometric indices".

Photographs in Figures 1 and 2 (especially for Figure 1) are of poor quality and does not give a real impression of plant morphology. Either replace with photographs giving clearer impression or discard them.

Lines 137–146, used abbreviations need to be given in full on first mention. 

Line 151, use "concentration" instead of "content" for total "phenolics" (instead of "phenols"), also for anthocyanins.

Discussion

The main problem is that Daily Light Integral has not been calculated in the present study, therefore, any comparison with studies using this parameter is relatively useless. 

Do not use author-centric style of narration, starting sentences with citation of author names. This is very distracting for the readers. Use problem-oriented style of narration instead. Introduce some logical structure how to compare the obtained results with facts from literature. More attention needs to be drawn to antioxidant activity as an important quality characteristic of vegetables.

Do not use redundant phrases "world researchers", "according to the literature", "the researchers".

Use past tense when referring to the own results.

Materials and methods

Source of seed material?

I am very much surprised by given value of PPFD in natural light, which is 80 μmol m–2 s–1. How this value was obtained? It does not seem to be right. Actual PPFD is changing on hourly and daily basis, and, in general, increases from the beginning of May towards the end of the May. Were actual PPFD measurements performed also for additional lighting treatments? What measuring device was used?

The actual manufacturer and brand of lamps used needs to be given. What were their spectral characteristics?

Line 277, how it was determined that plants "needed" water? How "similar" substrate moisture (not humidity) was measured and maintained? What type of water was used?

Brand of fertilizer needs to be given. Concentration and doses of fertilizer need to be mentioned.

Line 280, use "morphological parameters" instead of "biometric indices".

Is it meant that five plants were used for all morphological measurements, not only for leaf area?

Line 281, it is not becoming clear how representative plants were selected for analysis. If it was not randomly, then the procedure of selection needs to be described. 

Lines 287–290, how three plants were chosen from the five representative plants for analysis? Or these were different plants?

Conclusions

These are rather trivial, consider using more scientifically sounding text. Not clear what is meant by "internal plants quality", try to rephrase. "Cultivation costs" were not calculated, by the way.

Author Response

Response to Reviewer 3 Comments

We thank the reviewer for valuable comments on the manuscript. We are pleased to inform the reviewer that all suggestions have been considered and all requested edits have been made. The critical evaluation of the manuscript raised the following specific issues:

Title

Use "activity" instead of "response". 

Do not use "antioxidant response" also in other places of the manuscript.

Answer:

Changed lines: 3, 85, 148.

Abstract

Abstract is unbalanced, as results are mentioned only in a form of a short conclusion.

Rephrase the first sentence, as, for example, "Light and nutrients are among the most important factors for sustained plant production in agriculture".

Answer:

Changed line 10.

Line 11, use "the European Green Seal strategy". Start the sentence with "As" and finish with using "less energy-demanding growing systems".

Answer: Changed 11-12 lines.

Line 14, use "activity" instead of "response".

Changed.

Line 16, use "supplementing natural light with lighting at two photon flux densities".

Changed

Line 17–18, use "in each light regime".

Changed.

Line 19, use "with a complex fertilizer".

Changed.

Line 19–21, instead of a last sentence, describe the main results in two sentences and give a conclusion in a third.

 Changed.

Introduction

Lines 25–26, why the third factor is added to the two mentioned in the Abstract? It is rather confusing. Use the same construction as in the Abstract. 

Changed

Line 30, do not introduce abbreviation CEA, it is not further used.

Changed

Line 32–35, it is not becoming clear why the concept of daylight integral (or rather daily light integral) was introduced, as it is not included in the results of this study. No reference is given.

Changed see lines: 98-99, 115-116, 137-138, 271-274.

Lines 36–37, to avoid the impression that the study is important only for Lithuania, mention northern regions instead. Here, it is not light intensity, what is given, but rather daily light integral levels.

Line 41, use "the European Green Deal strategy".

Added, see lines 36-39.

Line 46–47, addressing "activity of the antioxidant system" (as an aspect of vegetable quality) needs to be explained in detail in a separate paragraph after describing light and nutrient effects on plant growth and production. 

Changed, see line 56,

Line 48–50, this thought is rather trivial.

Line 50–51, why only nitrogen and phosphorus are stressed out, there are number of other elements essential for plant growth?

Depending on the fertilizer used by the growers, it was chosen to study the composition of NPK fertilizer.

Line 56–57, there is no confirmation in a form of a reference that basil and lettuce are the most popular vegetables in the world. It is advised to omit specific details in this and the following paragraph on basil and lettuce cultivation and to transfer these to discussion. Instead, give general information on choice of model species and the main characteristics in respect to demands for light and nutrients, and describe general scientific problem on need to balance additional light and fertilization in respect to natural light conditions of the particular region for particular crop species.

Changed: see lines 57-60.

Results

In several places, the present tense is used instead of the past tense, consider changing all to the past tense (lines 107, 115).

Changed, see lines 110, 118.

Line 87, use "additional lighting" instead of "PPFD". Also, in other places, try to use "increasing PPFD" instead of simple "PPFD".

Line 95, use "morphological parameters" instead of "biometric indices".

Photographs in Figures 1 and 2 (especially for Figure 1) are of poor quality and does not give a real impression of plant morphology. Either replace with photographs giving clearer impression or discard them.

Lines 137–146, used abbreviations need to be given in full on first mention. 

Line 151, use "concentration" instead of "content" for total "phenolics" (instead of "phenols"), also for anthocyanins.

Changed, see lines: 95, 98, 155, 172-173.

Discussion

The main problem is that Daily Light Integral has not been calculated in the present study, therefore, any comparison with studies using this parameter is relatively useless. 

Do not use author-centric style of narration, starting sentences with citation of author names. This is very distracting for the readers. Use problem-oriented style of narration instead. Introduce some logical structure how to compare the obtained results with facts from literature. More attention needs to be drawn to antioxidant activity as an important quality characteristic of vegetables.

Do not use redundant phrases "world researchers", "according to the literature", "the researchers".

Use past tense when referring to the own results.

Answer: Thank you for your consideration. We have taken note and corrected according to your requirements. 

Materials and methods

Source of seed material?

All seeds were purchased from the national seed company "SĖKLOS"

I am very much surprised by given value of PPFD in natural light, which is 80 μmol m–2 s–1. How this value was obtained? It does not seem to be right. Actual PPFD is changing on hourly and daily basis, and, in general, increases from the beginning of May towards the end of the May. Were actual PPFD measurements performed also for additional lighting treatments? What measuring device was used?

We have written clarifications in the methodological part, in the lines: 273-276.

The actual manufacturer and brand of lamps used needs to be given. What were their spectral characteristics?

Answer:

PPFD was measured every two hours for three days - sunny, overcast and moderately cloudy in the greenhouse - and presented as an average.

Yes, the intensity of daylight was observed and the lighting was adjusted accordingly.

The lamp manufacturer was mentioned in the methodology; however, we supplemented the specifications of the LEDs.

See in line 277-278.

Line 277, how it was determined that plants "needed" water? How "similar" substrate moisture (not humidity) was measured and maintained? What type of water was used?

Answer: 

Substrate moisture was not measured, all plants were watered with equal amounts of water on the same day and hour. 

Added water quality indicators in the lines 284-285. (RO-type water, pH - 6.8, electrical conductivity 0,01 S m−1)

Brand of fertilizer needs to be given. Concentration and doses of fertilizer need to be mentioned.

Answer:

Fertilizer manufacturers are indicated in the methodology in 286-287 lines, as well as the ratio of elements in parentheses. See in line 287.

Line 280, use "morphological parameters" instead of "biometric indices".

Changed. 

Is it meant that five plants were used for all morphological measurements, not only for leaf area?

Answer:

Yes, we first weighed the plants, then measured their leaf area and put them to dry and weighed the biomass after drying.

Line 281, it is not becoming clear how representative plants were selected for analysis. If it was not randomly, then the procedure of selection needs to be described. 

Answer:

Plants were selected randomized, but some of the plants that may have had an edge effect were isolated.

Lines 287–290, how three plants were chosen from the five representative plants for analysis? Or these were different plants?

 Answer:

An additional three plants were selected for biochemical analysis in the same manner as for the evaluation of morphological parameters.

Conclusions

These are rather trivial, consider using more scientifically sounding text. Not clear what is meant by "internal plants quality", try to rephrase. "Cultivation costs" were not calculated, by the way.

Answer:

The findings were corrected in lines 362-367.

Round 2

Reviewer 3 Report

Not all suggestions have been properly addressed. There is no reason to give explanation for the reviewer, these need to be given in the text for the sake of clarity. 

Sorry for my typo in the comment "Line 11, use "the European Green Seal strategy"", it was meant to be "Green Deal".

The following are my comments from the previous review that have not been properly addressed in the manuscript.

Introduction

Line 30, do not introduce abbreviation CEA, it is further used only once.

Line 32–35, it is not becoming clear why the concept of daylight integral (or rather daily light integral) was introduced, as it is not included in the results of this study. No reference is given.

Line 41, use "the European Green Deal strategy".

Line 46–47, addressing "activity of the antioxidant system" (as an aspect of vegetable quality) needs to be explained in detail in a separate paragraph after describing light and nutrient effects on plant growth and production. 

Line 48–50, this thought is rather trivial.

Line 56–57, there is no confirmation in a form of a reference that basil and lettuce are the most popular vegetables in the world. It is advised to omit specific details in this and the following paragraph on basil and lettuce cultivation and to transfer these to discussion. Instead, give general information on choice of model species and the main characteristics in respect to demands for light and nutrients, and describe general scientific problem on need to balance additional light and fertilization in respect to natural light conditions of the particular region for particular crop species.

Results

Photographs in Figures 1 and 2 (especially for Figure 1) are of poor quality and does not give a real impression of plant morphology. Either replace with photographs giving clearer impression or discard them.

Lines 137–146, used abbreviations need to be given in full on first mention. 

Discussion

Do not use author-centric style of narration, starting sentences with citation of author names. This is very distracting for the readers. Use problem-oriented style of narration instead. Introduce some logical structure how to compare the obtained results with facts from literature. More attention needs to be drawn to antioxidant activity as an important quality characteristic of vegetables.

Materials and methods

Source of seed material?

Brand of fertilizer needs to be given. Concentration and doses of fertilizer need to be mentioned.

Lines 287–290, how three plants were chosen from the five representative plants for analysis? Or these were different plants?

Author Response

Response to Reviewer 3 Comments

Thank you, for your valuable comments. We believe we have corrected everything to your requirements.

Introduction

Line 30, do not introduce abbreviation CEA, it is further used only once.

Answer: Abbreviation removed

Line 32–35, it is not becoming clear why the concept of daylight integral (or rather daily light integral) was introduced, as it is not included in the results of this study. No reference is given.

Answer: We included DLI in the results section to explain the value of PPFD. See the lines 100-101, 113-115, 119-120, 129-131, 145-146, 157-158, 179-180, 188-198, 205-214.

Line 41, use "the European Green Deal strategy".

Corrected.

Line 46–47, addressing "activity of the antioxidant system" (as an aspect of vegetable quality) needs to be explained in detail in a separate paragraph after describing light and nutrient effects on plant growth and production.

 Answer: See lines 56-65.

Line 48–50, this thought is rather trivial.

Answer: Thank you for your opinion,the sentences have been changed.

Line 56–57, there is no confirmation in a form of a reference that basil and lettuce are the most popular vegetables in the world. It is advised to omit specific details in this and the following paragraph on basil and lettuce cultivation and to transfer these to discussion. Instead, give general information on choice of model species and the main characteristics in respect to demands for light and nutrients, and describe general scientific problem on need to balance additional light and fertilization in respect to natural light conditions of the particular region for particular crop species.

Answer: Changed in lines 66-68, and postponed to the beginning of the discussion as recommended. 

Results

Photographs in Figures 1 and 2 (especially for Figure 1) are of poor quality and does not give a real impression of plant morphology. Either replace with photographs giving clearer impression or discard them.

Answer: We don't have better quality photos of basil; would it be good to leave only the photo of the lettuce or better remove it all?

Lines 137–146, used abbreviations need to be given in full on first mention.

Changed in line 135. We believe that the abbreviations DPPH, ABTS, and FRAP are clearly explained in the methodology and it is not necessary to rewrite the full name of the radicals.

Discussion

Do not use author-centric style of narration, starting sentences with citation of author names. This is very distracting for the readers. Use problem-oriented style of narration instead. Introduce some logical structure how to compare the obtained results with facts from literature. More attention needs to be drawn to antioxidant activity as an important quality characteristic of vegetables.

Answer: the discussion has been corrected in lines: 220-223-234-238, 257-260-269, 297-298.

Materials and methods

Source of seed material?

Answer: See lines 314-315. (“Agrofirma SĖKLOS”, Vilnius, Lithuania)

Brand of fertilizer needs to be given. Concentration and doses of fertilizer need to be mentioned.

Corrected: See lines 330-332. (the other half of the plants were fertilized with complex NPK 3-1-3 fertilizers (Terra Grow, Plagron, Netherlands). 1 mL of fertilizer concentrate was diluted with 200 mL of water.)

 Lines 287–290, how three plants were chosen from the five representative plants for analysis? Or these were different plants?

Answer: They were different plants.